# Non-Contact Activity Monitoring Using a Multi-Axial Inertial Measurement Unit in Animal Husbandry

**DOI:** 10.3390/s22124367

**Published:** 2022-06-09

**Authors:** Pieter Try, Marion Gebhard

**Affiliations:** Group of Sensors and Actuators, Department of Electrical Engineering and Applied Sciences, Westphalian University of Applied Sciences, 45897 Gelsenkirchen, Germany; pieter.try@w-hs.de

**Keywords:** inertial measurement unit, activity classification, structural vibrations, machine learning, data fusion

## Abstract

In this work, a novel method is presented for non-contact non-invasive physical activity monitoring, which utilizes a multi-axial inertial measurement unit (IMU) to measure activity-induced structural vibrations in multiple axes. The method is demonstrated in monitoring the activity of a mouse in a husbandry cage, where activity is classified as resting, stationary activity and locomotion. In this setup, the IMU is mounted in the center of the underside of the cage floor where vibrations are measured as accelerations and angular rates in the X-, Y- and Z-axis. The ground truth of activity is provided by a camera mounted in the cage lid. This setup is used to record 27.67 h of IMU data and ground truth activity labels. A classification model is trained with 16.17 h of data which amounts to 3880 data points. Each data point contains eleven features, calculated from the X-, Y- and Z-axis accelerometer data. The method achieves over 90% accuracy in classifying activity versus non-activity. Activity is monitored continuously over more than a day and clearly depicts the nocturnal behavior of the inhabitant. The impact of this work is a powerful method to assess activity which enables automatic health evaluation and optimization of workflows for improved animal wellbeing.

## 1. Introduction

The monitoring of physical activity is a widespread tool to evaluate physical health in humans and animals alike. Physical activity is a complex multi-dimensional behavior that is characterized by frequency, duration, intensity and mode [1,2]. Typical metrics to quantify activity are locomotion (e.g., steps and traveled distance) and heart rate. The state of the art is wearable devices which contain various sensors to obtain objective measurements of activity [3,4]. However, wearable devices have some disadvantages that make them inconvenient and even unsuitable in some applications. For example, elderly people can forget to put the wearable device on and may be discouraged by the inconvenience of frequent charging. In addition, wearables are unsuitable for target groups such as small animals due to the relative weight of the wearable device, which has a strong negative impact. For these target groups, non-contact activity monitoring methods are employed which use cameras [5] and other sensing methods [6,7,8].

A major use case for non-contact activity monitoring is to evaluate health and wellbeing in the husbandry of mice. Mice husbandry, e.g., at the Institute for Laboratory Animal Science Aachen (VTK) which houses over 20,000 mice, is a large scale operation which requires a large workforce. Here, non-contact activity monitoring is used for behavioral studies and to ensure wellbeing of the animals. Sensing systems employed in this use case are cameras [9,10,11], radar sensors [12,13], capacitive sensing arrays [14] and arrays of radio-frequency identification (RFID) scanners [15,16]. However, these systems are only used on a small scale for studies and are not widespread in the husbandry where most animals reside. This is due to disadvantages such as high computation cost for camera systems, large sensing devices, invasive components (such as RFID-chips) and poor scalability.

Promising non-contact methods for activity monitoring have been presented previously [6,7,8,17,18,19,20,21], which are based on measurements of structural vibrations. These methods demonstrate how various activities and non-activity are detected in different use cases based on structural vibrations using various computational methods. State of the art in vibration sensing is the use of geophones, e.g., the SM-24 Geophone [22] which is used in most works mentioned above. Geophones have a common frequency bandwidth of 10 to 250 Hz, very low noise and high sensitivity which makes them suitable to measure small vibrations [22,23]. However, commercial multi-axial inertial measurement units (IMUs) have improved significantly in sensing performance and cost. One reason is the widespread adoption of micro-electro-mechanical system (MEMS) motion sensing devices in consumer electronics. A comparison of MEMS accelerometers and geophones is performed in a study [24,25]. In this study a single axis MEMS-based accelerometer for seismic applications is compared to several geophones. The results show that a MEMS-based accelerometer for seismic applications is competitive to geophones. However, as geophones are dedicated vibration sensing devices their performance in terms of noise and sensitivity is superior to commercial multi-axial IMU which are most often used for motion sensing. On the other hand, MEMS IMU have superior frequency bandwidth and integrate multiple sensor types in a highly integrated system on a chip (SOC). For example, the widespread 9-axis IMU or sometimes called magneto-inertial measurement unit (MIMU) hosts three triads of orthogonally oriented accelerometers, angular rate sensors and magnetometers in a package with a size of 3 × 3 × 1 ${\mathrm{mm}}^{3}$.

Looking at the state of the art, there is a lack of scalable non-contact methods for activity monitoring in animal husbandry. Such a method would allow for large scale automated activity and health assessment, which represents a major benefit for both animal welfare and caregiver workload. Furthermore, it is evident that multi-axial IMU are not yet state of the art for vibration measurement, despite the clear advantages of these sensors. Therefore, in this work, we propose a novel non-contact method for continuous activity monitoring, which uses a single commercial multi-axial IMU to measure the structural vibrations of a husbandry cage. Based on the cage vibrations, activity of the cage inhabitant is estimated and classified as resting, stationary activity and locomotion. For this, a setup is developed, which consists of an multi-axial IMU and a reference system. These are used to monitor the activity of a mouse inside a commercially available husbandry cage. The IMU sensor is mounted on the outside of the cage in the center of the cage floor. For reference of the activity of the inhabitant, a camera-based reference system is used. It uses a camera with a fish eye lens mounted in the cage lid and sophisticated algorithms for markerless motion tracking. Activity classes representing common activity patterns are defined based on the horizontal movement of the inhabitant. For activity classification based on structural vibration data, eleven features are extracted from multiple sensors of the IMU and used to train a supervised machine learning model. Training and evaluation is performed using 16.17 h of data.

This paper presents the proof of concept for a sensitive and selective activity classification method based on structural vibrations measured with an multi-axial IMU SOC. The results clearly show that IMU are suitable for measuring activity-induced structural vibrations with very small amplitudes. Is is shown that various cage vibrations in the three spatial directions are measured by the triads of orthogonally oriented sensors of the IMU SOC. Furthermore, it is shown that these vibrations contain non-redundant information that is fused to achieve increased activity classification accuracy. The evaluation further shows very good accuracy in identifying resting phases versus phases with activity. The activity of horizontal locomotion is identified in about half of the cases, which is detected based on unique vibrations induced by foot steps. Finally, it is demonstrated how the proposed method is used to monitor the activity over a period of 26.67 h. The results clearly show expected activity patterns such as the activity cycle of a nocturnal creature.

## 2. Concept, Setup and Reference

In this work, a proof of concept is presented for activity monitoring based on structural vibrations measured with a multi-axial IMU. Two major questions are explored for this proof of concept. First, IMUs are assessed for their performance in measuring structural vibrations as well as the activity-related information contained in it. Furthermore, metrics for measuring activity are assessed and classes of activity defined. For this, data of cage vibrations in the scenario of mice husbandry are acquired and analyzed. Afterwards, a method is researched to classify activity based on structural vibrations that are measured by the IMU. This method is developed based on observations from the acquired data.

In this section, the setup for data acquisition is presented as well as the initial analysis of IMU data with regards to activity-induced vibrations. Furthermore, the metrics are described by which activity is measured in this work. Afterwards, in Section 4, a method is presented to classify physical activity based on IMU measurements, which is built on the findings of this section.

### 2.1. Concept and Application Scenario

Physical activity monitoring is a valuable tool to evaluate health because of the correlation between health and physical activity. Generally speaking, physical activity is a complex behavior that is described by many characteristics. The commonly used metrics for activity are the amount of motion such as steps and travelled distance.

The movements that are performed during activity, apply dynamic forces to surrounding objects (e.g., the floor, a chair or a cage) which cause mechanical vibrations. Impulsive forces are mainly generated by localized impacts such as steps or an object dropping on the floor. These impulses create rapidly decaying vibrations which contain frequency components of the natural frequency of the excited object. Vibrations of objects on which activity takes place, for example vibrations of the floor or a husbandry cage, are referred to as structural vibrations. The structural vibrations created during different activities exhibit activity-dependent characteristics. For example during locomotion, short impulse-induced vibrations will occur at regular intervals corresponding to the frequency of the steps. In addition, the characteristics of vibrations vary due to the different strength and duration of impulsive forces. By measuring and analyzing structural vibrations for these characteristics, it is possible to identify the activity that is performed.

In mice husbandry, animals are held in rectangular plastic containers, shown in Figure 1 and Figure 2. These containers, consist of a cage tub and a cage lid. The cage tub is filled with one to two centimeters of bedding which is made from wood chips or similar material. The bedding provides a soft floor and absorbs moisture to improve hygienics. Two primary ventilation concepts are used, active ventilation and passive ventilation. In active ventilation, cages of type individually ventilated cage (IVC) are stored in racks which have an active ventilation system pumping fresh air through each cage individually. Passive ventilation relies on cutouts with air filters on the cage lid which provide fresh air. The mice held in large scale husbandry have a weight of approx. 25 g and typically inhabit cages in groups of up to six. However, this work will focus on a scenario with one mouse per cage.

In this use case scenario, a mouse is the subject which performs physical activity in a husbandry cage which is the surrounding object. During different activities, impulsive forces are applied to the cage by the inhabitant, which causes structural vibrations. Locomotion, digging and vertical stretching are some activities that are expected to cause structural vibrations. The forces generated during activity are very small due to the small size and weight of about 25 g. Correspondingly, the amplitude of vibrations is also small. In addition, activity induced forces are transferred to the cage indirectly through the soft bedding which dampens and decreases the amplitudes.

In this work, activity of the mouse is estimated by measuring the structural vibrations of the cage with a 9-axis IMU SOC as shown in Figure 1 and Figure 2. The proposed sensor integrates nine sensors in a single SOC and measures acceleration, angular rate and magnetic fields in the three spatial directions X, Y and Z. In this application, data of the magnetometer is omitted. The individual sensors are layered on top of each other in an integrated SOC using MEMS manufacturing technics. The complete package has a size of 3 × 3 × 1 ${\mathrm{mm}}^{3}$. Although IMUs are better described as sensor modules, in this work IMUs will be referred to as a sensor. When mounted to the cage floor, the IMU measures the vibrations of the cage at the mounting point as acceleration and angular rate in the three axis X, Y and Z. An advantage of the small dimensions of the IMU is the minimal influence on the natural frequencies of the cage. The acceleration measurements depict linear deflection and motion while the angular rate measurements depict twisting and rotational oscillations. The structural vibrations measured by the IMU are expected to exhibit different properties in the three spatial directions due to the geometry of the cage. In Z-axis a membrane-like vibration of the cage floor is expected. In X- and Y-axis vibration is expected to originate from oscillation of the whole cage, because these axes are aligned parallel to the plane of the cage floor. For concise monitoring and depiction of activity, metrics and classes have to be defined to describe different activity patterns of the inhabitant. This is described in Section 3.1.

The proposed method involves sensitive measurement of small cage vibrations to extract activity-specific information for activity classification. For this reason, it is of high interest to increase the signal to noise ratio due to the relatively large noise of the IMU, when compared to geophone sensors. However, the inherent noise of the IMU is relatively uniform in the frequency domain. Therefore, it is proposed to filter and extract activity-induced vibrations in the frequency domain where useful signals can be differentiated from the noise floor. In addition, the multiple information sources of the multi-axial vibration measurement are fused to achieve improved activity classification.

### 2.2. Setup and Measurements

The goal of the measurements is to collect IMU measurements of structural vibrations of a husbandry cage in the scenario that a mouse in the cage performs its regular activities undisturbed. For this, a measurement setup is used which simulates the environment found in regular husbandry and includes a camera reference system. It is shown in Figure 2 and used to acquire ground truth of physical activity using camera images. It is described in detail in Section 2.3. The setup consists of an unmodified Zoonlab HRC500 IVC [26], a cage lid with cutouts for the camera system, the camera reference system and the ICM20948 9-axis IMU from STMicroelectronics [27]. The IMU is mounted on the outside of the cage floor as shown in Figure 1. The camera reference system is mounted in the lid and provides a top-view of the inside of the cage. The top-view video images are later used to estimate physical activity with an algorithm, which serves as the ground truth for training and evaluation of algorithms. The cage is placed on a dedicated aluminum stand. The raw data streams of the IMU and camera are recorded and synchronized by a raspberry pi 4 using a program written in Python. In the measurements two setups, shown in Figure 2, were used in parallel to increase the amount of data collected.
Figure 2Figure shows the two measurement setups employed at the Institute for Laboratory Animal Science Aachen on the left. One the right the ICM20948 [27] is shown with a 2 Euro coin as reference for size.
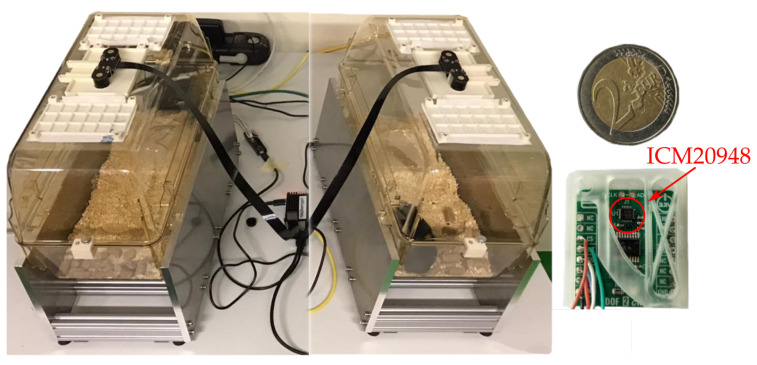


The ideal mounting position of the sensor, meaning a place with maximum vibration amplitude, was determined using finite element method (FEM) analysis of the empty cage. The result indicates that the cage floor exhibits a membrane-like oscillation when excited by an impulsive force. The center of the cage floor is the anti-node of this oscillation and has the highest amplitude of deflection. In addition, the FEM predicts a 1st harmonic natural frequency at 130 Hz. It should be mentioned that the exact material properties of the cage are not known and that the bedding and other objects inside the cage are neglected. Therefore, a lower natural frequency is expected in the real world scenario. Based on these results, the center of the cage floor is chosen as the mounting position of the IMU sensor. In order to prevent interference with the inhabitant, the sensor is mounted on the outside under the cage. The X-axis of the IMU is aligned to the short side, the Y-axis to the long side and the Z-axis perpendicular to the cage floor. The sensor is mounted using thin double sided tape of the brand tesa^®^, which is allows for non-permanent attachment of the sensor. Lab tests showed that vibration measurements are uninfluenced when using this tape for mounting. The IMU accelerometer data are obtained with a sampling rate of 4500 Hz (3 db cutoff at 1248 Hz) and a resolution of 61.04 μg/LSB. The manufacturer specifies a noise density of 190 $\mu g/\sqrt{\mathrm{Hz}}$ which amounts to a theoretical noise of 6.7 mg root mean square (RMS). The IMU gyroscope data are obtained with a sampling rate of 9000 Hz (3 db cutoff at 12,106 Hz) and a resolution of 7.63 mdps/LSB. The manufacturer specifies a noise density of 15 $\mathrm{mdps}/\sqrt{\mathrm{Hz}}$ which amounts to a noise of 1.65 dps RMS.

The measurements were carried out at the VTK [28]. Animal handling and cage preparations were performed by trained personal. Prior to the measurements, the cages were cleaned and fitted with new bedding (approx. 2 cm thick), food (45 g) and water (74 g). The measurements were performed in a room with certification for animal husbandry, which features automatic climate and light control for day night cycling. Four female mice, age 56 to 64 weeks of strain CD56/Black6 were used in the measurement. The measurements were performed as follows. The animals were placed inside the cages, the lid was closed and the data recordings were started. During measurements no humans were present in the room except for a short period to change out the cage inhabitants. Raw data of the IMU and the camera reference system are saved in packages of 10-min length. Data processing was performed after the measurement.

All in all, three datasets were collected: 27.67 h with one mouse (28 g) in the cage, 21.5 h with two mice (28 g and 27 g) and 9 h with three mice (28 g, 27 g, and 26 g). Each dataset consists of six IMU data streams at 4500 Hz (acceleration and gyro data each in the three axis) and a video stream at 1296 × 972 30 fps. However, only the first dataset with one mouse will be discussed in this work.

According to the ethical statement provided by the Landesamt für Umwelt-, Natur- und Verbraucherschutz of NRW Germany, these measurements are a pure observation without intervention in the habitat of the animals and therefore it is assessed not to be an animal experiment. Furthermore, food and water were provided in accordance to EU Directive (2010/63 EU) and the legal provisions of German Animal Welfare Act (TierSchG). Due to these circumstances ethical review and approval are waived.

### 2.3. Camera Reference System

It is necessary to determine the ground truth physical activity of the inhabitant for training of models, data analysis and evaluation. A camera-based reference system was developed for this, due to the availability of the deeplabcut framework for animal pose estimation [10] and because videos can be reviewed by humans. The system consisted of a PiCam NoIR-V1 camera module with a 160° field of view (FOV) lens. It was mounted in the lid of the cage which is modified with cutouts. The camera provided a top down view of the inside of the cage. A raspberry pi 4 was used to capture the camera images. The videos were processed afterwards, since deeplabcut is not real-time capable on the raspberry pi 4 hardware.

Physical activity can be described by many metrics and characteristics. In this work, velocity is chosen as a metric for activity, because it is a quantifiable measure of movement and can be reliably determined using deeplabcut by tracking the position of the inhabitant. The deeplabcut software package features pre-trained networks that are able to estimate the pose of various animals in various scenes. The pre-trained networks are retrained using a few manually annotated video frames. In this case, a deeplabcut network was retrained using 80 manually annotated video frames. The retrained deeplabcut network is then used to analyze a total of 16.17 h of video images. The deeplabcut network outputs the pose of the mouse as well as the likelihood of each label for each frame. Figure 3 shows a cropped video frame and the pose labels. The body shape of mice and the relative position of body parts varies drastically depending on the pose. In order to get a reliable velocity estimate, the head of the mice is chosen for velocity calculation, because it is rigidly coupled to the skeleton by the skull. Additionally, the head leads the motion of the body in most activities. The head velocity, in the following referred to as velocity, is calculated based on the position of the two ears and the headbase. The headbase is a point which lies between the ears. First the absolute velocity of each point is calculated separately using the position difference between two consecutive frames. Afterwards, the smallest of the three values is extracted for velocity estimation. This approach eliminates sudden velocity spikes that are caused by misclassification of individual points, which falsely suggest large movements. However, it leads to underestimation of velocity in rotational movements when velocity is different for the three points. However, in this case underestimation is more preferred than overestimation.

In order to determine activity classes based on velocity, seven clips of ten minutes each were manually labeled on a frame to frame basis using the activity labels: walking, digging, stretching, rotating, grooming and resting. These hand labels were chosen, because they are distinguishable from a top down view. Using these manually obtained activity labels, velocity-based activity classes are defined to establish the ground truth of physical activity.

## 3. Activity Classes and Their Correlation to IMU Measurements

The data acquired using the measurement setup described in Section 2.2 holds important information that: (a) is used to define a metric for objective activity estimation in our use case and (b) shows correlation of certain cage vibrations and activities. These results are presented in the following.

### 3.1. Definition of Camera-Based Activity Classes

The analysis of the deeplabcut data shows that the position of the body parts right ear, left ear and headbase are estimated most reliably based on their average likelihood. Therefore, it is confirmed that these points are suitable for reliable velocity estimation.

The Figure 4 shows two graphs with four plots containing activity and velocity data of one of the aforementioned seven clips which were manually annotated with activity labels. These plots show ten minutes of mouse activity with (A) the raw velocity values as explained in Section 2.3 (B) the raw velocity values averaged over an interval of 15 s (C) the activity labels by visual inspection and (D) the velocity-based activity labels.

When looking at the activity labels obtained by visual inspection (C) shown in Figure 4, it is observed that activities such as walking, digging and grooming are rarely performed continuously over a longer period of time. Instead, these activities are alternated in short intervals of a few seconds. This can be seen in the latter half of the plot where the activities walking, stretching and rotating are quickly switched between. The only activity that is performed over longer periods is resting which is depicted in the first half of the charts. This type of behavior is also observed in the other six clips which were manually inspected. From these observations, it is concluded that the activity classes resting, grooming, rotating, stretching, digging and walking, which were defined for manual annotation, are too detailed and unrepresentative of typical activity patterns. Therefore, three new activity classes are proposed which summarize certain individual activities to describe the most common activity patterns. These classes are used as the ground truth of physical activity in the rest of this work. Furthermore, it is determined that activity classification is not reliable on a frame to frame basis. The reason for this is that the activity patterns only become apparent over longer periods of time. Therefore, it is proposed to evaluate activity based on intervals with a timeframe of 15 s. The 15 s length is chosen based on observations that show a typical minimum activity duration of about 15 s.

In order to achieve a reliable ground truth for activity classification, it is proposed to use the velocity of the mouse, as the metric for classification. The velocity is obtained from pose estimation using deeplabcut and is shown as (A) in Figure 4. For classification of an 15 s interval, the average velocity (B) is calculated from the raw velocity data (A). The class is then determined with thresholds which are derived from correlations between the velocity and the common activity patterns seen in the activity labels obtained by visual inspection. This correlation can be seen in Figure 4 when comparing the plots of the two graphs. A detailed description of the activity classes and thresholds is given in the following. For reference, the mouse observed here has a length of approx. 200 px.

**Resting** In phases where activity is manually labeled as resting, the mouse exhibits a particular lack of motion in a sleep-like state which is clearly distinguishable from all other activities. Therefore, a velocity-based activity class is defined for the activity resting. When looking at the velocity of several timeframes which only contain resting, an average velocity of 0.45 px/s with a standard deviation of 0.04 px/s is observed. Based on this observation, the velocity-based label *resting* is assigned to all timeframes with an average velocity lower than 1 px/s. This means that timeframes with an average velocity above 1 px/s most certainly contain activity.**Locomotion** Locomotion is characterized by walking or a behavior where walking, digging, grooming, stretching and rotating are mixed up. It can range from slow walking to hopping. Timeframes during locomotion activities show an average velocity of 115.22 px/s with a standard deviation of 43.0 px/s. Based on this, timeframes with an average velocity of over 50 px/s are classified as locomotion. Due to the large standard deviation caused by the wide range of locomotion types, some misclassification is expected, for example during very slow walking.**Stationary Activity** The class stationary activity contains all activities which are performed on the spot such as grooming, digging, rotation and stretching. During timeframes with the aforementioned activities but without walking, an average velocity of 20.73 px/s with a standard deviation of 17.0 px/s is observed. Based on this, all timeframes with average velocities between 1 px/s and 50 px/s are considered stationary activity. The velocity distribution during stationary activity lies between the other classes. However, some overlap to locomotion may occur which is a source of misclassification.

The velocity-based activity labels as defined above are depicted as (D) in Figure 4. In this graph it can be seen that these activity classes have good correlation with the patterns exhibited in the manually obtained activity labels. Therefore it is considered to be a reliable ground truth of physical activity.

Apart from that, there are less frequent cases of activities which are atypical for their classes. For example, sometimes the mouse lifts and moves the water filled jellypack, which involves another type of motion typically not observed during stationary activity. These cases are not represented in the activity classes.

### 3.2. Correlation of IMU Data and Activity

Based on the three activity classes defined in the previous section, characteristics of cage vibrations measured with the multi-axial IMU are analyzed with regards to different activities. The cage vibrations, respectively the IMU data, are analyzed in the frequency and time domain using Fourier and short Fourier transformation. The Figure 5 illustrates correlations by showing synchronized plots containing activity labels and accelerometer data in the frequency domain over a time of ten minutes.

The Figure 5 shows four plots of a ten minute excerpt which depict the physical activity and the spectrogram of the accelerometer data. The spectrograms show the short time Fourier transformation of each of the three accelerometer axes. The Fourier coefficients are color coded, where blue represents a weak amplitude and yellow represents a strong amplitude. This figure serves to visualize correlations between the occurrence of vibrations in the three axes at certain frequencies and physical activity. With this visualization, multiple ten minute excerpts of data were analyzed and the observations are described in the following.

#### 3.2.1. Data of the Accelerometer

In the X-axis, which corresponds to vibrations along the short horizontal side of the cage, vibrations are observed at a frequency of 25 Hz while activities take place. During resting phases, the 25 Hz vibration has a very low amplitude which is indistinguishable from the noise floor in some cases. This can be seen in the first half of the graphs in Figure 5. In some instances when the inhabitant moves more strongly than usual, vibrations with higher frequency components are observed in a range of 100 Hz. This can be seen at the minute 8.5 and 9. Based on these observations, it is concluded that activity-induced vibrations most likely appear in a small frequency band around 25 Hz.

In the Y-axis vibrations with high amplitude are observed in the range of under 10 Hz. However, these have no apparent correlation with physical activity. Vibrations with weak amplitude in the range of 75 Hz to 80 Hz are infrequently observed during locomotion and stationary activity. During resting, these are indistinguishable from the noise floor. Based on these observations, activity-induced vibrations most likely appear in a range of 75 Hz to 80 Hz with small amplitude.

In the Z-axis, which lies perpendicular to the cage floor, a weak permanent vibration at 100 Hz persists regardless of the activity of the inhabitant. During locomotion and stationary activity, vibrations with frequency components in a range of 80 Hz to 250 Hz are observed. Most of the energy is concentrated around 120 Hz. These appear frequently during locomotion and are less common during stationary activity. This observation is made on several occasions including the clip shown in Figure 5. In some instances during locomotion or stationary activity, the inhabitant shows uncommon behavior with strong movements. For example, the mouse lifts and drops the jellypack during stationary activity which is not represented in the activity labels and has to be identified by manually viewing the video recordings. While this behavior occurs, vibrations with frequency components of 500 Hz to 1000 Hz are observed. These can be seen in Figure 5 where they are marked with the grey triangles. From these observations it is concluded that activity-induced vibrations commonly appear in a range of 50 Hz to 250 Hz.

In some instances not shown in Figure 5, high amplitude broadband vibrations are measured in all sensor axis. Velocity-based activity labels and manual reviewing of video material do not show any correlation to the behavior of the inhabitant.

In separate tests, measurements were made to evaluate the base sensors noise of the ICM20948 IMU. The results show a RMS noise of 6.6 mg, 6.5 mg and 6.4 mg for the X, Y and Z-axis of the accelerometer which lie within the margin of error of the value of 6.7 mg specified by the manufacturer.

#### 3.2.2. Data of the Gyroscope

In viewing the angular rate sensor data with regards to the velocity-based activity labels, activity-induced vibrations above the noise floor were not observed. Consequently, no activity related information could be obtained from the angular rate sensor of the IMU used in the measurement.

In separate tests, measurements were made to evaluate the base sensors noise of the ICM20948 IMU. The data of the gyroscope shows a RMS noise of 1.278 dps, 1.991 dps and 1.691 dps for the X, Y and Z axis which is slightly higher than the value of 1.65 dps specified by the manufacturer.

## 4. Method for IMU-Based Activity Classification

The result of the activity-induced vibration analysis, described in Section 3.2, suggests a strong correlation between the occurrence of certain signals in the three axis of the accelerometer data and the activity of the cage inhabitant. With these results, a feature set and a classification model are developed to classify the physical activity of the inhabitant only using data of the IMU. Classification is performed of 15 s intervals as previously defined in Section 3.1. A multi-class supervised approach is chosen based on the activity classes of the reference method defined previously. The classification model is trained using the Matlab Classification Learner app.

The feature set consists of statistical and frequency-domain features of the three IMU accelerometer axis. Features in the frequency domain are calculated using fast Fourier transform (FFT) algorithm and defined based on the findings of Section 3.2. The features are calculated over the 15 s timeframe and are listed below:

The dataset for training and testing is derived from 16.17 h of IMU measurements of cage vibrations and camera-based ground truth of physical activity. For obtaining the dataset, first the continuous data stream is divided into sections with a timeframe of 15 s. Each section is processed individually and represents one data point. A data point consists of two elements: (a) the 11 IMU-based features (b) the associated velocity-based activity label which represents the ground truth.

The whole dataset consists of 3880 data points. 1000 data points (26%) are randomly selected and withhold for testing. The remaining 2880 data data points (74%) are used for training and validation of classification models. After evaluation and testing of the classification model, a final classifier is trained using all available data points. Multiple classification models are automatically trained and validated using the Matlab Classification Learner app, including decision trees, discriminant analysis, support vector machines, logistic regression, nearest neighbors, naive Bayes, kernel approximation, ensemble, and neural network classification. Five-fold cross-validation is used for validation during training.

### 4.1. Method for Continuous Activity Monitoring

The continuous activity monitoring of the inhabitant of the husbandry cage is performed with the classification algorithm described above by analyzing consecutive intervals of the continuous IMU data stream. The continuous data stream is divided into sections with a duration of 15 s. The time at the beginning of each section is recorded in order to assign the activity classification to a point in time. Then, activity is estimated as described above by calculating the features and applying the classification model. The result is continuous monitoring of activity with a time resolution of 15 s.

## 5. Results on IMU-Based Activity Classification

All in all, 3880 data points (74% of the dataset) containing the features listed in Table 1 were used in the training of multiple classification models. Another 1000 data points (25.7%) were used for testing. The training dataset contains 1105 (38.37%) samples of the activity class resting, 1317 (45.73%) samples of the class stationary activity and 458 (15.9%) samples of class locomotion. The test dataset contains 321 (32.1%) samples of the activity class resting, 510 (51.0%) samples of the class stationary activity and 169 (16.9%) samples of class locomotion. The results of the evaluations are presented in the following.

### 5.1. Best Fit Classification Model

In multiple training runs involving all classification models available in the Matlab Classification Learner app, validation accuracy between 68% and 80% are observed. The optimizable ensemble classification model achieved the best results in validation and test accuracy persistently. Consequently, it was deemed best fit to this classification problem and used in the following evaluations.

### 5.2. Classification Accuracy

The optimizable ensemble classification model achieves a validation accuracy of 82.3% and a test accuracy of 79.8% when all features are used. The confusion matrix is presented in Figure 6. In testing, the class resting has the highest true positive rate (TPR) of 94.1% with a PPV of 84.8%. The class stationary activity ranks second and achieves a TPR of 82% with a positive predictive value (PPV) of 80.2%. The class locomotion achieves a TPR of 46.2% with a PPV of 63.4%.

When comparing the results of the validation during training and the results achieved on unseen test data, the total accuracy differs by 2.5%. In the classes resting and stationary activity the TPR between validation and testing differs by less than a percent. In the class locomotion a difference of about 10% is observed.

### 5.3. Results of Using Features from Different Vibration Axes

In order to evaluate the information contained in the data of each accelerometer axis, five optimizable ensemble classification models were trained with different features. The results are presented in Table 2.

The best accuracy is achieved by the classification models which use all features and features from the X- and Y-axis accelerometer data combined. They are followed by the model using Z-axis features, then the model using X-axis features and at last the model using Y-axis features.

### 5.4. Final Classification Model

The final classification model is trained using all 3880 available data points. All features are used in the training of this optimizable ensemble classification model. The validation result is shown in Figure 7.

Validation shows an accuracy of 81.8%. The class resting has the highest TPR of 92.8% with a PPV of 88.3%. The class stationary activity ranks second and achieves a TPR of 82.2% with a PPV of 80.2%. The class locomotion shows a TPR of 55.8% with a PPV of 68.8%.

### 5.5. Application of Continuous Activity Monitoring

Using the method described in Section 4.1, continuous activity monitoring is demonstrated on the measurements obtained at the VTK. The final classification model, which is trained using 16.17 h of data, is used to analyze the 27.67 h of data collected during the measurements at the VTK. The 16.17 h of data used for training are contained in the 27.67 h of data. For continuous activity monitoring, the data are divided into sections with a timeframe of 15 s length. Each timeframe is classified individually. The output of this operation is an array with 6640 elements. Each element contains the point of time and the activity label. The output array is plotted in Figure 8.

Figure 8 shows the result of the proposed method for continuous activity monitoring over a timespan from 9:23 to 15:30 of the next day. The figure shows two plots. The upper plot contains each individual activity classification which has a duration of 15 s. The lower graph shows a summarized representation where activity is described by the percentage of occurrence of each label over an interval of 30 min. The activity classes are color coded. The color black stands for resting, gray for stationary activity and red for locomotion. The mouse is placed in the measurement cage a few minutes prior to the start of the measurement period. The room where the cages were placed was only visited in the period which is marked by the circled number two. Otherwise no humans intervened during the measurements. The light of the room is dimmed between 19:00 and 7:00 to simulate the night which is marked by the bar with the circled number 3. Between 5:30 and 8:20 the sensor detached from the cage which is clearly indicated in the IMU data. As a result, no IMU data are collected in this time span. At around 8:20, the sensor is reattached and the data acquisition is continued.

The graphs shown in Figure 8 clearly depict patterns of activity which change over time. In the time period between 9:30 and 11:30 a lot of locomotion is observed at the beginning which decreases more or less linearly. This is followed by a phase dominated by resting which lasts until 19:00. Between 19:00 and 5:40 higher activity levels are seen with increased locomotion of around 10% to 20% and little resting phases as well as a lot of stationary activity at over 50%. The time period from 8:23 to 16:00 is dominated by resting which is interrupted by the occasional stationary activity and locomotion. In total, the activity is monitored over 27.67 h. The inhabitant is observed to rest for a total of 13.09 h. In sum, stationary activity is observed over 11.44 h and locomotion over 3.13 h

## 6. Discussion

In this work, a proof of concept is presented for a method to continuously monitor activity based on measurements of structural vibrations using a multi-axial IMU. In order to proof the feasibility of the concept, two major questions were explored. These are the performance of multi-axial MEMS IMU in measuring structural vibrations as well as physical activity monitoring based on data of an IMU mounted on a husbandry cage. The results are discussed in the following chapter.

### 6.1. Mechanics of Activity-Induced Cage Vibrations

The measurements of the IMU depict the vibration of the cage at the mounting point in the three spatial directions. The vibrations in the three axis exhibit different characteristics in frequency, amplitude, decay rate and occurrence. In the analysis, various patterns are evident that correlate with the physical activity of the inhabitant. Based on the rather high frequency of these vibrations, it is evident that these vibrations do not directly correspond to periodic impulsive forces such as footsteps or grooming. It is hypothesized that these vibrations are natural vibrations of the cage which are induced by non-periodic impulsive forces generated during activity.

Based on this hypothesis, vibrations measured in the Z-axis of the IMU, which exhibit a typical frequency of 120 Hz, most likely correspond to the membrane-like natural oscillation of the cage floor. This theory is supported by the FEM result which predicts oscillation of the cage floor at 130 Hz. The lower frequency seen in the measurements is explained by the added mass of the bedding and the mouse inside the cage, which are neglected in the simulation.

The vibrations measured in the X- and Y-axis of the IMU most likely correspond to an oscillatory movement of the cage in the horizontal plane. These can either be deformations or movement of the cage in the suspension. Nevertheless, the vibration characteristics such as frequency and amplitude are determined by the geometry and material properties and are different in the X- and Y-axis.

In summary, the analysis of the IMU data clearly shows three dimensional activity-induced structural vibrations of the husbandry cage. The observed activity-induced vibrations exhibit frequencies in the range of 0 Hz to 250 Hz with higher frequency vibrations generated by strong movements in the range of 500 Hz to 1000 Hz. From these results it is concluded that commercial MEMS IMU are capable of measuring activity-induced structural vibrations for classification of activity. Furthermore, these sensors have the necessary frequency bandwidth to measure the natural oscillations of the cage which are induced by activity.

### 6.2. Quality of Reference Activity Labels

The camera reference system classifies activity based on the two dimensional velocity of the mouse inside the cage which is used as ground truth to classify activity based on cage vibrations measured with an IMU. Three activity classes are defined for velocity-based classification which represent the most frequent behavior patterns resting, stationary activity and locomotion. When comparing data of the velocity-based activity classes to the activity labels obtained by visual inspection, it is observed that the velocity-based activity classes are a good representation of mice behavior. However, the three classes only represent an approximation of activity, due to the complex nature of physical activity.

In comparison, the velocity-based activity class resting is well distinguished from the other classes due to the narrow distribution of velocities during resting phases. Therefore, good separation of non-activity (resting) and activity (the other two classes) is achieved. However, due to the wide distribution of velocities in the classes stationary activity and locomotion, some uncertainty remains in the distinction of the two classes. Among other things, this is due to the fact that the intensity and quantity of movement varies drastically in the defined classes. Due to these circumstances, the velocity by itself is not conclusive enough for correct classification of the behavior. In order to improve the activity estimate, additional metrics such as distance traveled in a timeframe is proposed to be used.

### 6.3. Performance of Activity Classification Using IMU-Based Cage Vibrations

In the evaluation of the classification model, similar results were achieved in the validation of the training dataset and testing of the test dataset. It is concluded that generalization error of the classification model is small which makes it suitable to analyze previously unseen data.

The results presented in Section 5.3 provide insight into the activity-related information contained in the vibrations measured in each axis. All in all, the classification model which uses all features achieves the best results. The classifier which uses data of the X- and Z-axis achieves similar results. A significant difference could not be determined. The classification model, which only uses features from the vibrations in the Y-axis, achieves poor accuracy and PPV of around 50%. These results suggest activity-related information contained in the data of the Y-axis is small. The classification models, that use data of the X- or the Z-axis exclusively, achieve a similar accuracy. However, significant differences are seen when comparing the TPR and PPV of the models in different classes. From these results, it is concluded that the information contained in the data of different axes are non-redundant and enable better overall classification when information is fused for classification.

Furthermore, the evaluation of the models shows that classification performance differs significantly between the classes. In the following, classification performance of the individual activity classes is discussed:**Resting**:Very good performance is achieved in the classification of resting with a TPR of about 94% and a PPV of about 85%. The results from Section 5.3 show that resting is best classified when features from the X- and Z-axis are used in combination. This is supported by the analysis from Section 3.2 which shows that activity-induced vibrations have a particularly low amplitude and occurrence during resting. The observations show that the mouse sleeps or is in a sleep-like state during resting phases. In this state, the mouse almost does not move which in turn generates almost no vibrations. As a result, resting can be differentiated well even from stationary activity.**Stationary Activity**:The class stationary activity is classified reliably with a TPR of 82.0% and a PPV of 80.2%. Stationary activity is misclassified as resting in about 10% of cases. This is in part explained by phases with little movement that sometimes occur during time periods with stationary activity. In other cases, stationary activity is misclassified as locomotion which is in part due to strong movement on the spot which induces vibrations in the Z-axis of the cage floor. One exemplary case is when the mouse interacts with the jellypack and throws it to the ground which generates strong vibrations in the Z-axis. Table 2 shows that the model only using X-axis features achieves better TPR in classifying stationary activity by about 11%, compared to the model only using Z-axis features. This is attributed to the observations, that vibrations in the X-axis occur very reliably during any sort of activity, while vibrations in Z-axis mostly occur in locomotion activity shows that stationary activity has a higher TPR in the model only using X-axis**Locomotion**:The class locomotion is classified unreliably with a 56% TPR and 63.4% PPV. However, when locomotion is misclassified, it is almost exclusively misclassified as stationary activity. Locomotion is only misclassified as resting in 1% to 3% of cases. This suggests that the information extracted from the IMU data is not sufficient to reliably distinguish between stationary activity and locomotion. The results presented in Table 2 show that the model using Z-axis features has a 23.7% better TPR than the model using X-axis data. This result shows that features from the Z-axis are the best predictors of locomotion. This is supported by the observations in Section 3.2 which indicate that vibrations in the Z-axis are measured more frequently during locomotion than during the other two classes. The vibrations in Z-axis are thought to be membrane-like vibrations of the cage floor that are induced by impulsive forces generated by steps. However, the analysis also shows that these vibrations appear unreliably during locomotion which is a contributing factor to the unreliable classification of locomotion. It is likely that step induced vibrations occur more frequently than measured but exhibit an amplitude that is indistinguishable from the sensor noise.

### 6.4. Continuous Activity Monitoring

The method for continuous activity monitoring described in Section 5.5 is used on the IMU measurements of over one and a quarter day made at the VTK. It demonstrates the continuous monitoring of the activity of a mouse inside a husbandry cage. The monitoring data, shown in Figure 8, depicts clear trends and behavioral patterns that are discussed in the following.

In the beginning of the measurement period, the data shows high activity levels with little to no resting and a lot of locomotion which decreases almost linearly in the timespan between 9:23 and 11:30. This behavior is attributed to an acclimatization phase due to being put into a freshly cleaned and replenished cage. Furthermore, data suggests a nocturnal behavior where higher levels of activity are observed in the evening hours between 19:00 and 8:30 and lower levels of activity during the day, especially between 8:30 and 15:30 of the second day. These observations coincide with the expected nocturnal behavior of mice and suggests that general behavioral patterns can be deduced from the activity monitoring data.

Over the whole measurement period of 27.67 h, monitoring data shows a total resting time of 13.09 h and a total active time of 14.57 h. This results suggests a sleep duration of 47% per 24 h, although this value is somewhat imprecise due to the data loss between 5:30 and 8:20 in the morning. However, the observed resting time coincides with literature [29] which suggests that mice spend about 40% to 50% of a day sleeping or in a sleep-wake state.

In conclusion, the data derived from the proposed continuous activity monitoring method shows clear behavioral patterns that coincides with the expected behavior described in the literature. Therefore, it is concluded that this method is able to monitor the activity of the mouse to such an extent that reliable statements about the general behavior can be made. In addition, the method also enables the estimation of activity duration such as time spend sleeping per day. However, for confirmation if activity durations can be extracted reliably, more data must be collected for evaluation.

## 7. Conclusions

In this work, a novel method is presented for physical activity classification and monitoring, which uses a multi-axial IMU SOC to measure structural vibrations in multiple axes. The method is demonstrated in the use case of continuous activity monitoring of a mouse in a shoe-box-style husbandry cage. In this use case, an IMU is mounted in the center of the underside of the cage floor to measure structural vibrations in the three spatial directions. The vibrations are measured in three axes as acceleration and angular rate. A reference system is developed which uses top-view video recordings to classify physical activity into the classes resting, stationary activity and locomotion. These classes are defined using observations obtained by visual inspection of videos and reflect the most common physical activity patterns. The reference system is assembled in the lid of a husbandry cage. This setup is used to obtain 27.67 h of IMU data of cage vibrations as well as synchronized video recordings. Based on this data, different activity-induced vibration modes of the cage are identified as well as their correlation to certain activity patterns. From this, eleven features are defined as predictors of activity classes which contain and fuse information from the X-, Y- and Z-axis accelerometer data. These features are used to train a classification model with a dataset containing 16.17 h of IMU cage vibrations and ground truth activity labels. The resulting classification model achieves a TPR of 94% in classifying resting, 82% in classifying stationary activity and 46% in classifying locomotion on previously unseen data which was withheld from training. Using this classification model, continuous activity monitoring is demonstrated on the 27.67 h of collected IMU data which spans a period of 9:30 a.m. to 15:30 p.m. of the next day. The results of the proposed monitoring method clearly illustrate the behavioral patterns of the mouse which coincide with the expected behavior described in literature. Additionally, the method is able to measure the duration of activities such as sleeping which is useful for the assessment of health.

In summary, a novel method is demonstrated which enables continuous activity monitoring of a mouse in a husbandry cage using a non-contact non-invasive sensing method. This sensing method, allows for the monitoring of physical activity in regular husbandry cages where mice spend most of their lives. In addition, hardware requirements only consist of an IMU, which is mounted on the outside of the floor of the cage. Furthermore, it is shown that more information is extracted and better activity estimation achieved, when structural vibrations are measured in multiple axes and the data fused. As a result, excellent results are achieved in the distinction between activity (the classes stationary activity and locomotion) and non-activity (the class resting). The impact of this work is a powerful method to continuously monitor physical activity using a low-cost IMU. This method for activity monitoring provides a tool to assess the activity of an animal inside its regular habitat on a large scale, which opens up many possibilities to automatically monitor wellbeing and create optimized work flows.

## 8. Future Work

Based on the promising results in this work, one goal of future work is to expand the application scenario to the activity monitoring of multiple inhabitants in one husbandry cage. Data of this scenario was previously collected which is described in Section 2.2.

As described in Section 6.3 it is likely that many activity induced vibrations in the Z-axis remain undetected due to sensor noise. In addition, no activity-induced signals are observed in the data of the angular rate sensor. For this reason, other commercial IMU with lower sensor noise are to be investigate for improved sensing of activity-induced structural vibrations. Furthermore, the use of magnetometers in a sensor system for sensing structural vibrations will be investigated due to the full market penetration of low cost 9-axis IMU in portable devices. These sensor modules contain nine orthogonally mounted triads of accelerometers, angular rate sensors and magnetometers.

In analyzing the results, there is evidence for cage vibrations in X- and Y-axis. It is beyond the scope of the work presented here, which is a proof of concept for activity monitoring, to discuss structural dynamics in depths. Therefore, extensive FEM modal analysis of the cage is planned, to identify oscillation modes of the cage and their correlation to activity-induced impulsive forces.

A major aspect of future works is research on algorithms, classification methods and sensors for more accurate activity monitoring. First, more sophisticated algorithms will be investigated to analyze the individual activity-induced structural vibration. For example, wavelet transformation-based filtering of time domain signals which will be used to detect individual activity-induced vibrations in the time domain. It is also investigated, how activity-related information can be obtained from the angular rate sensor and magnetometers, which will be fused to improve activity classification. In addition, more secure mounting solutions are developed to prevent detachment of the IMU from the cage. At last, due to their small size it will be investigated how multiple IMU can be integrated into a sensor system to achieve more accurate detection of activity-induced structural vibrations and in turn improve activity classification.

Lastly, the camera reference system is to be improved and made real-time capable in classifying activity using top-view video images. This enables longer measurement periods for evaluation and training purposes which were challenging previously due to the massive amount of video data produced.

## Figures and Tables

**Figure 1 sensors-22-04367-f001:**
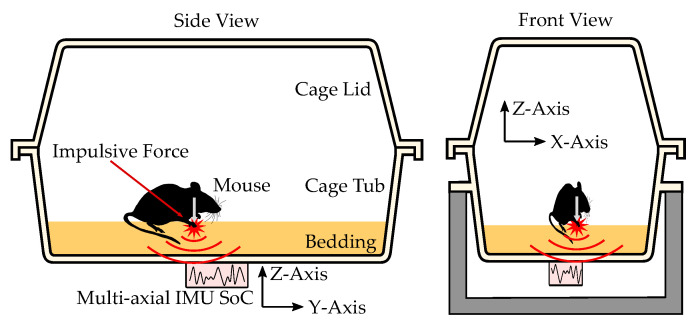
This drawing visually depicts the concept of the proposed method for activity detection and classification based on measurements of structural vibrations with an multi-axial inertial measurement unit.

**Figure 3 sensors-22-04367-f003:**
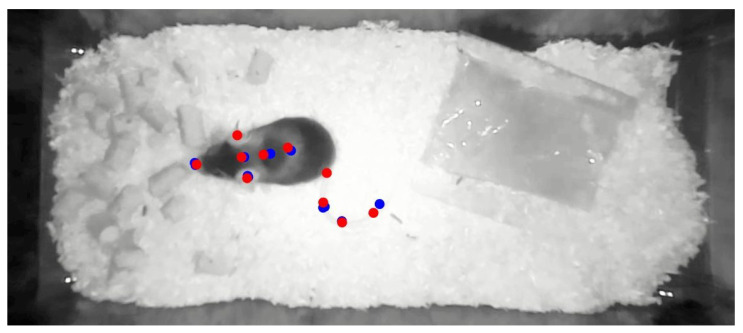
This figures shows a video frame of the camera reference system. The red dots illustrate manually set pose labels. The blue dots are estimates of a retrained deeplabcut network.

**Figure 4 sensors-22-04367-f004:**
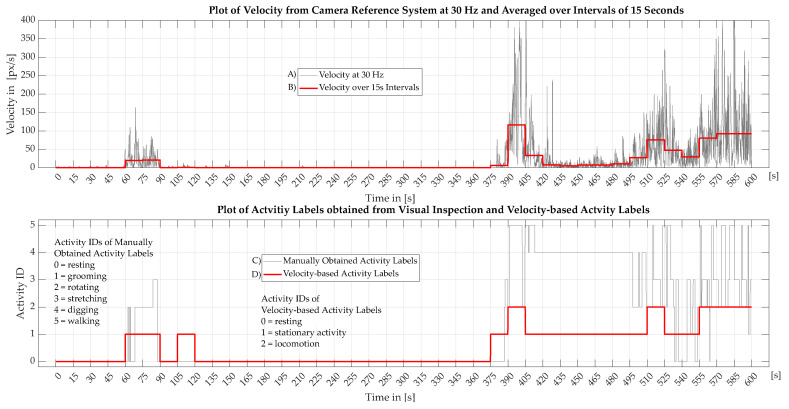
This figure shows two graphs with four plots containing various activity labels and the velocity of the mouse of a 10-min excerpt. The upper graph depicts the velocity of the mouse obtained from the camera reference system at 30 Hz as well as average values over intervals of 15 s. The lower graph illustrates activity labels from visual inspection of the camera images as well as velocity-based activity labels. The activity classes are depicted as activity IDs ranging from 0 to 5 for plotting purposes. The X-axis of the charts are synchronized.

**Figure 5 sensors-22-04367-f005:**
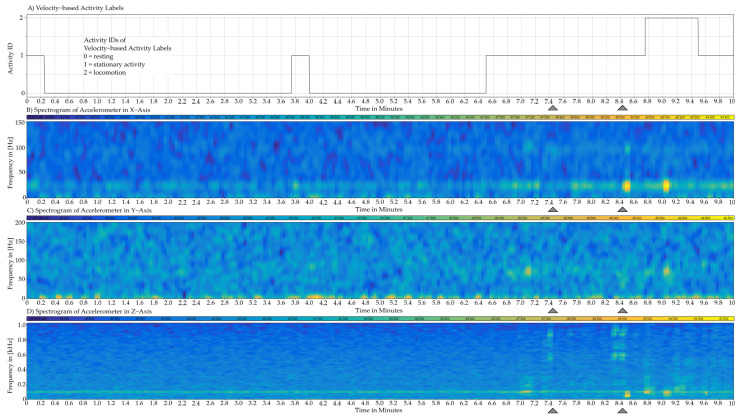
This figure shows four synchronized plots over a length of ten minutes which contain ground truth activity labels and the spectrogram of all accelerometer axis. The gray triangles mark two special case activities, that were observed by manually viewing the video recordings.

**Figure 6 sensors-22-04367-f006:**
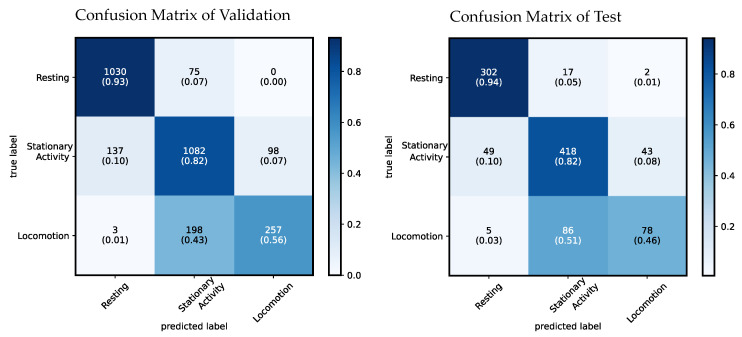
This figure shows the confusion matrix of an optimizable ensemble classification model which computes eleven features. The numbers in the boxes show the amount of samples. The numbers in the brackets below show the relative amount of samples. Validation results are displayed on the left and test results on the right.

**Figure 7 sensors-22-04367-f007:**
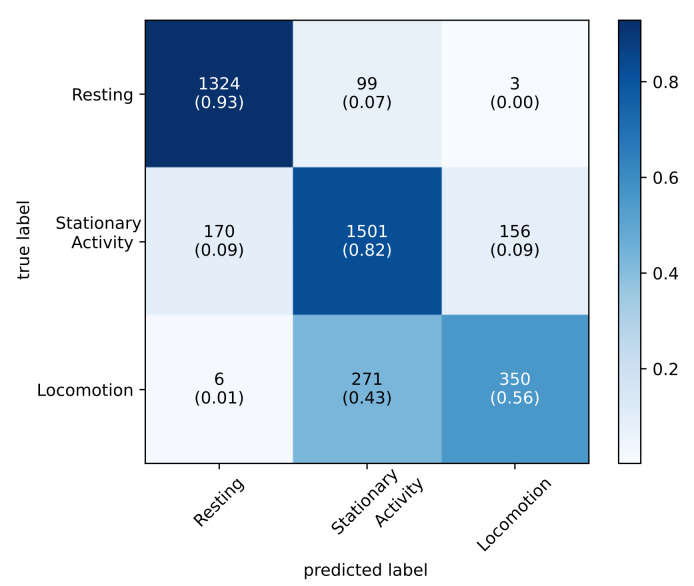
Confusion matrix of the final classification model using all available data points. The numbers in the boxes show the amount of samples. The numbers in brackets below show the relative amount of samples.

**Figure 8 sensors-22-04367-f008:**
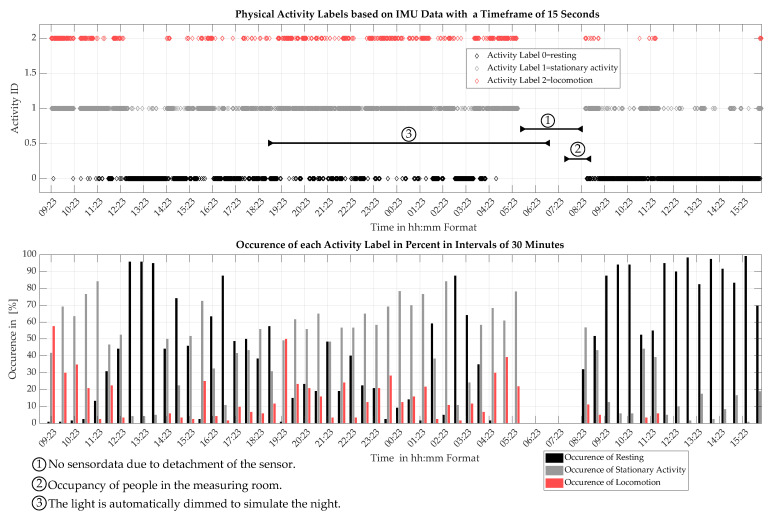
This figure shows the physical activity of the inhabitant of the husbandry cage over one and a quarter of a day. The activity labels are estimated from cage vibrations measured with an 9-axis IMU. The upper plot shows 6640 data points of activity labels which represent the activity patterns of 15 s intervals. The lower plot shows the occurrence of each activity in an interval of 30 min in percent. Special events are marked using bars.

**Table 1 sensors-22-04367-t001:** Feature set for classification of physical activity based on 3-axis accelerometer data of a cage mounted inertial measurement unit.

Number	Axis	Description
1	X	Relative average of FFT coefficients in the interval 23 Hz to 27 Hz
2	X	Absolute average of FFT coefficients in the interval 23 Hz to 27 Hz
3	X	RMS of time-domain signal
4	Y	Relative average of FFT coefficients in the interval 62 Hz to 80 Hz
5	Y	Absolute average of FFT coefficients in the interval 62 Hz to 80 Hz
6	Y	RMS of time-domain signal
7	Z	Relative average of FFT coefficients in the interval 80 Hz to 120 Hz
8	Z	Absolute average of FFT coefficients in the interval 80 Hz to 120 Hz
9	Z	Relative average of FFT coefficients in the interval 60 Hz to 180 Hz
10	Z	Absolute average of FFT coefficients in the interval 60 Hz to 180 Hz
11	Z	RMS of time-domain signal

**Table 2 sensors-22-04367-t002:** This table shows the test results of five ensemble classification models. The models were trained using different sets of features which is described in the left most column. TPR stand for true positive rate and PPV for positive predictive value.

Classes	Resting		Stationary Activity		Locomotion		Accuracy
	TPR	PPV	TPR	PPV	TPR	PPV	
All Axis	94.1%	84.8%	82.0%	80.2%	46.2%	63.4%	79.8%
X- And Z-Axis	93.5%	85.7%	83.5%	80.1%	46.2%	66.1%	80.4%
Only X-Axis	86.6%	84.5%	83.1%	71.7%	25.4%	53.8%	74.5%
Only Y-Axis	79.8%	48.6%	41.4%	59.1%	32.0%	46.6%	52.1%
Only Z-Axis	93.1%	77.1%	71.8%	79.2%	49.1%	55.3%	74.8%

## Data Availability

Data and algorithms presented in this work can be found here: https://github.com/pietertry/Non-contact-Activity-Monitoring-Using-a-Multi-Axial-Inertial-Measurement-Unit-in-Animal-Husbandry.git, accessed on 11 May 2022.

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
