# Peer review of "Non-Contact Activity Monitoring Using a Multi-Axial Inertial Measurement Unit in Animal Husbandry"

_sensors, 2022, doi:10.3390/s22124367_

Round 1
Reviewer 1 Report
In this paper, a novel method is presented for non-contact non-invasive physical activity monitoring, which utilized a multi-axial IMU to measure activity-induced structrural vibrations in multiple axes. This paper can be accepted after minor revision:
(1) The motivation and contribution should be given in introduction point by point;
(2) Fig. 5 is not clear enough, which should be re-plot.
Reviewer 2 Report
In the future work, it will be interesting to use Wavelet transform for De-noising of IMU output signals.
Reviewer 3 Report
The paper has a very good quality describing detailly all important parts of authors research work. Similar experiments with MEMS have been already provided with many authors. This paper concentrates on advanced technology and commercial low cost MEMS stressing authors selection of the measurement device using a serious analysis of competitive technology.
A novel method is demonstrated which enables continuous activity monitoring of a mouse in a husbandry cage using a non-contact non-invasive sensing method. This sensing method, allows for monitoring of physical activity in regular husbandry cages The selected commercial IMU is mounted on the outside of the floor of the cage. It is shown that more information is extracted and better activity estimation achieved, when structural vibrations are measured in multiple axes and the data fused. Good results are achieved in the distinction between activity (the classes stationary activity and locomotion) and non-activity (the class resting). Authors approach offers a powerful method to continuously monitor physical activity using a low-cost IMU. As experiments have got some limitations, authors plan to continue with development more precise measurement system including improving identification algorithms, classification methods, and camera reference system to made real-time capable in classifying activity using top-view video images.
References are serious, in total 29, with a few dated in period 2006-2008 describing basic theoretical principles and majority dated since 2015 to 2022.
